# Genome-Wide Identification of Long Non-coding RNA in Trifoliate Orange (*Poncirus trifoliata* (L.) Raf) Leaves in Response to Boron Deficiency

**DOI:** 10.3390/ijms20215419

**Published:** 2019-10-31

**Authors:** Gao-Feng Zhou, Li-Ping Zhang, Bi-Xian Li, Ou Sheng, Qing-Jiang Wei, Feng-Xian Yao, Guan Guan, Gui-Dong Liu

**Affiliations:** 1National Navel Orange Engineering Research Center, College of Navel Orange, Gannan Normal University, Ganzhou 341000, China; zhougaofeng428@163.com (G.-F.Z.); zhangliping0412@163.com (L.-P.Z.); libixian208@163.com (B.-X.L.); fengxianyao@aliyun.com (F.-X.Y.); guanguan_1985@aliyun.com (G.G.); 2Institute of Fruit Tree Research, Guangdong Academy of Agricultural Sciences, Guangzhou 510640, China; shengou6@yahoo.com; 3College of Agronomy, Jiangxi Agricultural University, Nanchang 330045, China; qingjiangwei520@163.com

**Keywords:** Citrus, boron deficiency, long non-coding RNAs, co-expression, plant hormone

## Abstract

Long non-coding RNAs (lncRNAs) play important roles in plant growth and stress responses. As a dominant abiotic stress factor in soil, boron (B) deficiency stress has impacted the growth and development of citrus in the red soil region of southern China. In the present work, we performed a genome-wide identification and characterization of lncRNAs in response to B deficiency stress in the leaves of trifoliate orange (*Poncirus trifoliata*), an important rootstock of citrus. A total of 2101 unique lncRNAs and 24,534 mRNAs were predicted. Quantitative real-time polymerase chain reaction (qRT-PCR) experiments were performed for a total of 16 random mRNAs and lncRNAs to validate their existence and expression patterns. Expression profiling of the leaves of trifoliate orange under B deficiency stress identified 729 up-regulated and 721 down-regulated lncRNAs, and 8419 up-regulated and 8395 down-regulated mRNAs. Further analysis showed that a total of 84 differentially expressed lncRNAs (DELs) were up-regulated and 31 were down-regulated, where the number of up-regulated DELs was 2.71-fold that of down-regulated. A similar trend was also observed in differentially expressed mRNAs (DEMs, 4.21-fold). Functional annotation of these DEMs was performed using Gene Ontology (GO) and the Kyoto Encyclopedia of Genes and Genomes (KEGG) pathway analyses, and the results demonstrated an enrichment of the categories of the biosynthesis of secondary metabolites (including phenylpropanoid biosynthesis/lignin biosynthesis), plant hormone signal transduction and the calcium signaling pathway. LncRNA target gene enrichment identified several target genes that were involved in plant hormones, and the expression of lncRNAs and their target genes was significantly influenced. Therefore, our results suggest that lncRNAs can regulate the metabolism and signal transduction of plant hormones, which play an important role in the responses of citrus plants to B deficiency stress. Co-expression network analysis indicated that 468 significantly differentially expressed genes may be potential targets of 90 lncRNAs, and a total of 838 matched lncRNA-mRNA pairs were identified. In summary, our data provides a rich resource of candidate lncRNAs and mRNAs, as well as their related pathways, thereby improving our understanding of the role of lncRNAs in response to B deficiency stress, and in symptom formation caused by B deficiency in the leaves of trifoliate orange.

## 1. Introduction

Boron (B) is an essential micronutrient for the growth and development of vascular plants [1]. It is widely known that B plays an important role in various metabolic processes, including plant phenolic metabolism, membrane integrity and function nitrate assimilation, along with cell wall synthesis and structure [2,3,4]. However, B deficiency is frequently observed, and is one of the major constraints to the production of 132 crops in many parts of the world [5]. Under B deficiency conditions, plant growth and development are significantly inhibited because B is essential to the structure and function of plant cell walls [6]. Citrus is an important fruit crop that also suffers from B deficiency stress in China and other countries [5,7]. Recent physiological and molecular studies on the effect of boron deficiency stress on citrus plants have focused upon the symptoms, growth and development, photosynthesis, nutrient absorption, transport and distribution and metabolism [8,9,10,11,12,13,14,15]. Those studies revealed that corky split vein was the typical symptom B deficiency stress in the leaves of citrus plants, which differs from those of other species, and its mechanism remains unclear. Previous studies have demonstrated that lignin metabolism is involved in the formation of symptoms in citrus plants under B deficiency conditions [12,13]; however, its mechanism is unclear.

Non-coding RNAs (ncRNAs) consist of three subgroups based on the length of nucleotides, namely, Non-coding RNAs (ncRNAs) (18–30 nt), medium-sized ncRNAs (31–200 nt) and long ncRNAs (lncRNAs) (> 200 nt) [16,17]. Similar to other ncRNAs, lncRNAs are functional RNAs with low protein-coding potential, except for *ENOD40*, which has been reported to encode small peptides [18]. Moreover, based on its genomic origins, lncRNAs are broadly divided into three types: Intronic lncRNAs (incRNAs), intergenic lncRNAs (lincRNAs), and antisense lncRNAs [19]. Coupled with the rapid development of genome-wide transcriptome sequencing technology, a large amount of lncRNAs responsive to biotic and abiotic stresses have been identified in plants. Previous researches have indicated that lncRNAs play a role in various important biological processes, including biotic and abiotic stress responses in plants [20,21,22]. By the end of 2015, PLNlncRbase has a total of 1,060 lncRNAs that are involved in abiotic or biotic stress responses in 43 plant species [23]. These lncRNAs respond to stresses using five different ways, namely: As ceRNAs, as sRNA precursors, antisense transcription, histone modification and DNA methylation [22]. Nutrient deficiency or toxicity is a common abiotic stress for vascular plants, and these stresses will trigger the expression of lncRNAs. Recently, several studies have shown that lncRNAs play a critical role in the regulation of gene expression in response to nutrient stress. For example, under phosphate (Pi) starvation conditions, lincRNAs (one type of lncRNAs) were systematically annotated in rice, and those involved in the biological response to Pi starvation were identified [24]. Similar studies were also conducted in other nutrient stresses and species, such as nitrogen (N) deficiency in populus [25], N and Pi starvation in rice [26], Pi starvation in *Arabidopsis thaliana* [27] and high B level in maize [28]. On the other hand, lncRNAs are considered potential regulators of plant responses to nutrient stress. For instance, overexpression of the lncRNA *IPS1* (*Induced by Phosphate Starvation 1*) results in reduced shoot P concentration in *Arabidopsis thaliana* [29]. Nonetheless, a few lncRNAs have been characterized to regulate plant responses or tolerance to B deficiency stress. In addition, the role of lncRNAs in response to B deficiency stress is poorly understood. Therefore, the present study performed a genome-wide identification and characterization of lncRNAs in response to B deficiency stress in the leaves of trifoliate orange (*Poncirus trifoliata*), an important rootstock of citrus. Our purpose was to investigate the role of lncRNAs in response to B deficiency stress in relation to the development of leaf symptoms.

## 2. Results

### 2.1. The Differential Performance of the Leaves under B Deficiency Stress

After 12 weeks of B deficiency treatment, the differential performance between B deficiency (BD) and control (CK) plants was observed in the leaves. Vein swelling and cracking (typical symptoms of citrus) was observed in the old leaves of BD after 12 weeks B deficiency stress. Meanwhile, leaf rolling without chlorosis was also found in the old leaves, but no symptoms were observed in the CK (Figure 1A,B). Anatomical assessment was performed on the malformed veins by paraffin sectioning, and the result showed that the epidermis was destroyed due to the proliferation of the vascular cylinder (Figure 1C,D). Further electron microscopy analysis demonstrated that the wall of the epidermis cells and xylem vessel were significantly thicker in BD (Figure 1E,G). The electron microscopy results also showed that many starch grains had accumulated in the leaf cells under B deficiency stress (Figure 1E). These results indicated that B deficiency can severely damage the vascular tissues and induce hypertrophy. Compared with CK, leaf B concentrations decreased, and the lignin content was significantly increased under B deficiency stress (Figure 1I,K).

### 2.2. Identification and Characterization of lncRNAs and mRNAs in the Trifoliate Orange Transcriptome

To investigate the lncRNAs in citrus under B deficiency stress, whole transcriptome strand-specific RNA sequencing was performed in the leaves of trifoliate orange seedlings treated by BD and CK. Initially, six high-throughput sequenced transcriptomes were generated, consisting of more than 0.67 billion clean reads, of which three were produced from the BD raw reads (BD_1: 113,804,668; BD_2: 100,898,426, and BD_3: 102,976,228, respectively), and the other three were produced from the CK raw reads (CK_1: 106,350,380, CK_2: 128,309,312, and CK_3: 119,745,274, respectively). These results indicate that more than 99% of the raw data were clean reads (Table 1). The clean reads rate and Q20% and Q30% results showed that the quality of the raw data met the requirements, and the GC separation phenomenon was not observed in the raw data (Table 1). The analysis reflected that our data were reliable.

As shown in the computational pipeline (Figure 2), all of the initial cDNA sequences with lengths <200 nt were eliminated, because lncRNAs should be longer than 200 nt. Further sequences were subjected to ORF prediction. ORFs of <120 amino acids were retained, and the remaining sequences were discarded. Finally, a total of 2101 putative lncRNAs and 24,534 mRNAs were identified in the genome of trifoliate orange using RNA-seq in the present study.

### 2.3. Genomic Organization of Trifoliate Orange lncRNAs According to the Clementine Reference Sequence

LncRNAs are shorter and possess fewer exons than protein-coding transcripts. Based on the genomic localization relative to clementine mandarin (*Citrus clementina*) known protein-coding genes, we analyzed the number of exons and the length distribution between lncRNAs and protein-coding transcripts. Figure 3A shows that 80.6% of lncRNAs ranged in length from 200 to 1000 nucleotides, and only 19.4% were longer than 1000 nucleotides. In contrast, 75.1% of the protein-coding transcripts were longer than 1000 nucleotides. Additionally, 60.2% of the lncRNAs consisted of zero or one exon, while more than 86.0% of the protein-coding transcripts had more than one exon (Figure 3B). Hence, most of the lncRNAs were shorter and had fewer exons relative to the protein-coding transcripts. According to the location relative to the nearby protein-coding genes, lncRNAs were classified into four types: Intergenic (48.18%), overlapping (44.55%), antisense (5.45%) and intronic/exon (1.82%). Most lncRNAs were located within intergenic regions (Figure 3C).

### 2.4. Differential Expression of mRNAs and lncRNAs in Response to B Deficiency

To validate the expression levels of mRNAs and lncRNAs in the leaves of trifoliate orange under B deficient stress, we randomly selected eight DELs and eight DEMs and quantified them using a quantitative real-time polymerase chain reaction (qRT-PCR) (Figure 4). The results showed that the expression patterns of specifically expressed DELs and DEMs in the sequencing and qRT-PCR results were mostly consistent, although the relative expression levels of all DEMs and DELs measured by RNA-Seq were greater than those by qRT-PCR.

Integrative analysis of genome-wide lncRNA and mRNA expression in the leaves of trifoliate orange under B deficiency stress were conducted in the present study. A total of 1547 significant differentially expressed mRNAs (DEMs) were identified (Appendix A). The number of up-regulated DEMs was significantly higher than that of down-regulated DEMs (1250 DEMs were up-regulated and 297 DEMs were down-regulated). The number of DEMs was 4.21-fold higher in up-regulated relative to down-regulated DEMs.

To identify the differentially expressed lncRNA (DELs) in the leaves of trifoliate orange subjected to B deficiency stress, the normalized expression (RPKM) of the lncRNAs was compared between the BD and CK groups. Total of 1,450 lncRNA were identified, and with |log_2_FC(BD/CK)| ≥ 2 as a cutoff, a total of 115 lncRNAs were significantly differentially expressed in response to B deficiency stress (Appendix A), including 84 up-regulated and 31 down-regulated lncRNAs, respectively (Figure 5). As shown in Figure 5, the expression patterns of DEMs and DELs showed similar trends under B deficiency conditions.

Volcano plots analysis was then applied for the direct identification of differences in lncRNAs and mRNAs in BD and CK (Figure 6A,B). Hierarchical clustering technique was used to separate the DELs and DEMs between BD and CK in term of the gene expression data (Appendix A).

### 2.5. GO and KEGG Enrichment Analysis of DEM and DEL Target Genes

To gain insights into the biological roles of DEMs in the leaves of trifoliate orange under B deficiency stress, all DEMs were assessed by Gene Ontology (GO) and the Kyoto Encyclopedia of Genes and Genomes (KEGG) pathway enrichment analyses. These DEMs were obviously responsive to the stimulus (GO: 0050896), chemical (GO: 0042221) and stress (GO: 0006950) of biological processes (Figure 7A). The DEMs were significantly enriched in the cell periphery (GO: 0071944), kinesin complex (GO:0005871) and apical plasma membrane (GO: 0016324) of cellular components (Figure 7B). In addition, the DEMs were significantly enriched in ATP-dependent microtubule motor activity (GO: 1990939), microtubule motor activity (GO: 0003777) and hydrolase activity, along with hydrolyzing *O*-glycosyl compounds (GO: 0004553) of molecular function (Figure 7C). Moreover, DEMs were significantly enriched in a series of KEGG pathways such as metabolic pathways (KEGG: ko01100), biosynthesis of secondary metabolites (KEGG: ko01110), carbon metabolism (KEGG: ko01200), phenylpropanoid biosynthesis (KEGG: ko00940) and glycolysis/gluconeogenesis (KEGG: ko00010) (Figure 7D).

GO enrichment analysis on the target genes of DELs was also carried out to investigate the potential functions of lncRNAs. As shown in Figure 8, there are 30 GO terms which were divided into three secondary classifications: 16 GO terms belong to the biological process category, 6 GO terms belong to the cellular component category, and 8 GO terms belong to the molecular function category. In this biological process category, many will target genes in response to stress (GO: 0006950) and in response to auxin (GO: 0009733). These results indicate that auxin may play a key role in response to B deficiency stress in the leaves of the trifoliate orange plant. The results also showed that many target genes were involved in cell membrane (GO: 0016020) and cell periphery (GO: 0071944) in the cellular component category. However, the count of target genes of the molecular function category is significantly less than those of the other two categories; in addition, the expression profiles of significantly differentially expressed target genes were also analyzed (Appendix A). A total of 157 target genes were up-regulated, but only five target genes were down-regulated under B deficiency stress. There are a lot of target genes involved the lignin metabolism, plant hormone metabolism and cell wall metabolism that were up-regulated under B deficiency stress. There are many transcription factors which were also found, such as the WRKY, MYB, bHLH and bZIP transcription factors. It is noteworthy that cytochrome P450 (Ciclev10031635m, as the target gene of lncRAN XLOC_011386) was significant up-regulated under B deficiency stress. These results indicated that all these target genes may play an important role in citrus in response to B deficiency stress.

### 2.6. LncRNA-mRNA Interaction Network Analysis

The relationship between lncRNA and protein-coding genes was visually displayed by interactive networks using Cytoscape software (3.6.1). The interactive network was composed of 558 network nodes and 838 connections between 90 lncRNA and 468 significantly differentially expressed target genes (Appendix A). Interestingly, all these lncRNA-mRNA pairs with the same expression trend under B deficiency conditions.

As shown in Appendix A, the lncRNA-mRNA interaction network analysis results indicated that one lncRNA could regulate many mRNAs (protein-coding genes), and in turn one mRNA could be regulated by many lncRNAs. In the Figure 9A the local network was constructed with lncRNAs, which top five nodes have were regarded as the key objects. LncRNA XLOC_002048 has 42 target protein-coding genes, lncRNA XLOC_025377, XLOC_001503, XLOC_011386 and XLOC_003172 interact with 33, 32, 32 and 30 target genes, respectively. 

Further study should be conducted on these lncRNAs. The expression of these target protein-coding genes was also analyzed, and all genes were up-regulated significantly under B deficiency stress (Figure 9B). Target genes with top five nodes were also analyzed, and the results show that Ciclev10014993m, Ciclev10004519m, Ciclev10006674m, Ciclev10019991m and Ciclev10022921m interacted with 9, 8, 7, 7 and 7 lncRNAs, respectively, and both target genes and lncRNAs are up-regulated under B deficiency stress (Figure 10). The gene description of the targets was shown in Appendix A.

### 2.7. LncRNA Involved in Plant Hormone Biosynthesis and Signal Transduction

GO enrichment analysis on target genes of differentially expressed lncRNAs under B deficiency stress indicated that many lncRNAs are involved in plant hormone biosynthesis and signal transduction. As shown in Appendix A, a total of 37 non-repetitive target genes were related to auxin (IAA) biosynthesis, transmembrane transporter activity, signal transduction, and so on. Similarly, there are eight non-repetitive target genes which were related to cytokinin (CTK), nine with gibberellin acid (GA), 37 with abscisic acid (ABA) and 11 with ethylene (ETH). The number of IAA- and ABA-related target genes was significantly higher than in CTK, GA and ETH.

After expression analysis, significantly up- or down-regulated expressed lncRNAs and their target genes were selected and constructed the co-expression network (Figure 11). A total of 13 lncRNAs and 12 target genes that were related to IAA were significantly up-regulated under B deficiency stress. Further lncRNA-target genes co-expression network analysis indicated that there were 25 nodes and 13 connections between these lncRNAs and target genes. For CTK, three lncRNAs and three target genes constructed three connections (Figure 11B). For GA, four lncRNAs and five target genes constructed five connections (Figure 11C). A total of 16 lncRNAs (14 up-regulated and 2 down-regulated) and 15 target genes (13 up-regulated and 2 down-regulated) related to ABA were selected and constructed a co-expression network with 16 connections (Figure 11D). For ETH, only six lncRNAs and six target genes were significantly up-regulated under B deficiency stress, and six connections were constructed between them (Figure 11E).

## 3. Discussion

Recently, accumulating evidences have indicated that lncRNAs play various critical roles in multiple biological processes in plants, but studies investigating the characteristics, expression patterns and potential functions of lncRNAs in plants in response to B deficiency stress are very limited. In our work, RNA-seq was used for lncRNAs analysis and three aspects were investigated: (1) The lncRNAs, as well as mRNAs, in response to B deficiency stress were identified; (2) their expression patterns under B deficiency stress; and (3) the effects of lncRNA on genes expression in BD. Our results provide new insights into the regulatory functions of lncRNAs in citrus in response to B deficiency stress, providing abundant information for further investigations on the molecular mechanism underlying citrus responses to B deficiency stress.

### 3.1. The Expression of lncRNAs and mRNAs involved in Several Metabolic Pathway are Altered in Response to B Deficiency Stress

B deficiency is involved in many metabolism processes, including lignin biosynthesis, photosynthesis, cell wall synthesis, IAA metabolism and phenolic metabolism [3,4,12]. In this study, functional annotation of DEMs was performed using GO and KEGG pathway analyses, and the results demonstrated that the biosynthesis of secondary metabolites (including phenylpropanoid biosynthesis/lignin biosynthesis), photosynthesis and the transport of photosynthetic production (including photosynthesis-antenna proteins, starch and sucrose metabolism, carbon fixation in photosynthetic organisms), and plant hormone signal transduction, play important roles in response to B deficiency stress (Figure 7).

Lignin biosynthesis is affected by B deficiency stress [30]. Our previous studies showed that the lignin biosynthesis pathway of citrus was influenced under the B deficiency condition [12,13]. Under B deficiency conditions, the genes encoding key enzymes in the lignin biosynthesis pathway were significantly up-regulated, such as phenylalanine-ammonium lyase (PAL), 4-coumarate: CoA ligase (4CL), cinnamoyl-CoA reductase (CCR) and peroxidase (POD). The higher expression of these genes will result in the accumulation of lignin. As shown in Figure 1, lignin content increased in BD. Therefore, our results agree with the previous conclusion that B deficiency has a significant influence on lignin biosynthesis in plants. Further studies demonstrated that the accumulation of lignin was deposited on the cell wall as observed by an electronic microscope and histochemical staining [12]. The heavily thickened cell walls may play an important role in symptom formation under B deficiency stress.

Previous research showed no direct evidence to demonstrate that B plays a role in photosynthesis [31]. However, several recent studies have indicated that B deficiency can significantly reduce the net photosynthetic rate of many citrus species [8,9,32,33]. Moreover, in this work, the starch and sucrose metabolism, which are the main photosynthetic products, was affected under B deficiency stress (Figure 7). Similar results were also reported in another citrus [8]. Electron microscopy showed that many starch grains accumulated in the leaf cells under B deficiency stress (Figure 1E). Accumulation of photosynthetic products could be one of the key factors leading to the decreased biomass of any plant subjected to B deficiency stress.

### 3.2. The role of Plant Hormones in Symptom Formation and Exacerbation

In this study, the lncRNAs target genes were investigated by GO enrichment analysis to understand their functions (Figure 8 and Figure 11; Appendix A). Recently, a number of previous researches showed that lncRNAs play an important role in the response of plant hormones to biotic and abiotic stress. For instance, GO pathway enrichment analysis of annotated genes showed that the differentially expressed lncRNAs were related to abscisic acid (ABA) and ethylene (ETH) biosynthesis and signal transduction in switchgrass under dehydration stress [34]. Other researchers found that the lncRNAs may be likely involved in regulating plant hormones pathway in response to drought stress in cotton such as cytokinin (CTK), gibberellin acid (GA), ETH and auxin (IAA) [35]. 

Plant hormones can adjust plant growth to environmental conditions such as nutrient availability, including B deficiency. Previous study on *Brassica napus* indicates that a variable B nutritional status causes coordinated changes in plant hormone metabolism as a prerequisite for an adjusted growth response [36]. As shown in Appendix A, based upon prediction, we obtained some lncRNAs, which may be the regulators of genes that control the expression of plant hormones in the leaves of trifoliate orange under B deficiency conditions. Finally, a total of 37, 8, 9, 37 and 11 non-repetitive target genes were related to IAA, CTK, GA, ABA and ETH, respectively. These target genes play important roles in their biosynthesis, transmembrane transporter activity, signal transduction, and so on.

Interestingly, the number of IAA- and ABA-related target genes was significantly higher than in CTK, GA and ETH. Previous studies have shown that B deprivation harms many physiological processes, including the IAA metabolism and others [2,37]. In recent years, IAA was demonstrated to be involved in the root growth and development of trifoliate orange under B deficiency conditions [13,38]. On the other hand, it has been proven that auxin can also control the development of leaf and vein [39]. In this study, a total of 18 GO pathways were related to auxin and GO: 0009733 has the highest number of genes (Appendix A). The target gene Ciclev10003896m (in GO:0060774), which is an auxin-mediated signaling pathway that is involved in phyllotactic patterning, was significantly up-regulated under B deficiency conditions (Figure 11A). Another target gene Ciclev10019991m (in GO: 0009850) has the second-highest log_2_FC(BD/CK) value (Figure 11A). Meanwhile, this target gene was also co-expressed with seven lncRNAs (Figure 10). Therefore, these genes may play very important roles in citrus in response to B deficiency stress by the auxin mediated signaling pathway.

In this work, seven GO pathways were related to CTK and three pairs of co-expression lncRNA-target genes were constructed (Figure 11B; Appendix A). Previous research showed that CTK can promote the differentiation of tracheary element [40], as well as promote the proliferation of cambial cells [41]. Further researches have shown that CTK plays a role in cell division and differentiation via signal transduction pathways [42,43]. Recent research on the corky split veins symptom of ‘Newhall’ navel orange (*Citrus sinensis* Osb.) has shown that genes associated with plant CTK signal transduction, cell division and vascular development were affected in corky split veins [13]. Thus, these studies indicated that the CTK signal transduction pathway may play a role in the symptom formation and exacerbation of citrus suffering from B starvation stress. Taken together, our results indicate that the symptoms presented in the leaves and veins of trifoliate orange, which were caused by B deficiency stress, might be mediated by plant hormone signal transduction pathways.

Interestingly, besides the plant hormone signal transduction pathway, the calcium (Ca^2+^) signaling pathway was also enriched by B deficiency stress as indicated by KEGG analysis (Figure 7D). As shown in Appendix A, the calcium-dependent protein kinase gene (Ciclev10008338m), which is the target gene of lncRNA XLOC_002224, was up-regulated in the leaves of trifoliate orange under B deficiency stress. Ca^2+^ is an important second messenger that plays a major role in plant responses to biotic and abiotic stress, such as nutrient deficiency stress. Previous researches have shown that Ca concentrations increase in citrus under B deficiency stress [11]. At the cellular level, B deficiency increases cytosolic Ca^2+^ levels in *Arabidopsis thaliana* roots and promotes tobacco BY-2 cells to take up more Ca^2+^ than control cells [44,45]. A recent study demonstrated that B deficiency increases cytosolic Ca^2+^ levels mainly via Ca^2+^ influx from the apoplasts in *Arabidopsis thaliana* roots [46]. At the molecular level, the expression of Ca^2+^-related genes in these *Arabidopsis thaliana* roots significantly increased under B deficiency conditions [45]. Therefore, based on the published articles, researchers propose and discuss that Ca^2+^ and Ca^2+^-related proteins-channels/transporters, sensor relays and sensor responders, might play major roles as intermediates in a transduction pathway triggered by B deprivation [47]. Our results agree with these hypotheses and demonstrate that Ca^2+^ signaling transduction plays an important role in the response of citrus to B deficiency stress.

### 3.3. The Mechanism of lncRNAs Response to B Deficiency Stress

Recently, numerous studies indicated that lncRNAs play a critical role in the regulation of gene expression in response to nutrient deficient or toxic stress in vascular plants [20,21,22]. For instance, N deficiency in populus [25], phosphate (Pi) starvation in rice [24,27], N and Pi starvation in rice [26] and excess B in maize [28]. Unfortunately, the underlying regulation mechanisms of lncRNA in response to B deficiency stress are unknown. Our results present the global characterization of lncRNAs and their potential target genes in response to B deficiency stress in citrus, which provides more information on B deficiency response mechanisms in citrus. Previous research indicated that lncRNA can cis- or trans-regulate the expression of target genes, which is the main regulation mechanism of lncRNA in plant. In this work, based on the analysis of the DEL-DEM co-expression networks, lncRNA XLOC_002048, XLOC_025377, XLOC_001503, XLOC_011386 and XLOC_03172 had high connectivity with DEMs in the leaves of trifoliate orange under B deficiency stress (Figure 9). XLOC_002048 was up-regulated by 8.11-fold in B deficient trifoliate orange, which was co-expressed with 42 DEM, and all the DEMs were up-regulated. Changes in the expression of DELs and DEMs changes in response to B deficiency, and could be an adaptive mechanism that contributes to the tolerance of plants to B deficiency stress. However, the roles of these DELs in relation to the response of plants to B deficiency remain unclear. Thus, the role of these lncRNAs should be further characterized to identify a direct regulation of genes associated with the responsive mechanism of B deficiency stress in citrus. For example, lncRNA IPS1 was overexpressed in *Arabidopsis thaliana*, in which P concentration was also significantly decreased in shoot [29]. Similar studies will be designed and carried out in citrus, as well as the further studies on the confirmation of their functions.

Additionally, most lncRNAs have been shown to regulate target genes expression directly through complementary binding to protein-coding transcripts in the form of cis-antisense, but they can also modulate transcription factors to regulate target genes expression indirectly [48,49]. In this work, many transcription factors were found in significantly differentially expressed target genes, such as the WRKY transcription factor, MYB transcription factor, bHLH transcription factor and bZIP transcription factor (Appendix A). The expressions of all these transcription factors were up-regulated under B deficiency stress. Our results indicated that the mechanism of lncRNAs response to B deficiency stress may exist via the transcription factor. Therefore, further studies should be focused on these lncRNAs which target genes belong to transcription factor family.

## 4. Materials and methods

### 4.1. Plant materials and B Deficiency Treatment

Seeds of trifoliate orange (*Poncirus trifoliata* (L.) Raf.) were used in this experiment. Seed germination, pre-cultured in soil and hydroponic culture, were performed according to Zhou et al. [12]. Plant hydroponic cultures were prepared with 1/2 strength Hoagland’s No. 2 nutrient solution, which contained 6 mM KNO_3_, 4 mM Ca(NO_3_)_2_, 1 mM NH_4_H_2_PO_4_, 2 mM MgSO_4_, 9 µM MnCl_2_, 0.8 µM ZnSO_4_, 0.3 µM CuSO_4_, 0.01 µM H_2_MoO_4_ and 50 µM Fe-EDTA [50]. Before B deficiency stress treatment, the pre-cultured plants in normal solution were transferred either to a new nutrient solution with 0.01 mg·L^−1^ B for B deficiency (BD) treatment or to a nutrient solution with 0.25 mg·L^−1^ B as the control (CK). After the B deficiency treatments for 12 weeks, leaves were sampled and immediately frozen in liquid N_2_ and stored at −80 °C until RNA extraction.

### 4.2. Determination of B and Lignin Concentration

After the B deficiency treatments for 12 weeks, the leaves of nine trifoliate orange plants per treatment were randomly harvested for B and lignin concentration determination. The fresh materials were placed into a forced air oven at 105 °C for 15 min, and then at 75 °C until constant weights were reached. 

All the dried samples then were ground into fine powder for the determination of their B concentration. 0.50 g of leaf sample were dry-ashed in a muffle furnace at 500 °C for 6 h, followed by dissolution in 0.1 N HCl, and B was determined with Inductively Coupled Plasma-Mass Spectrometry 7900 (ICP-MS7900; Agilent Technologies Inc., Santa Clara, USA). Lignin was extracted and measured by the method of Bruce and West [51].

### 4.3. Anatomical Analysis

Microscopy observation was performed on the main veins of trifoliate orange according the paraffin method previously described by Yang et al. [13]. For transmission electron microscopy, ultrathin sections were prepared using an ultramicrotome (MT-X; RMC), and the sections were thoroughly stained with aqueous 2% uranyl acetate for 10 min followed by lead citrate for 2 min. The sections were viewed with a JEM-1010 electron microscope (JEOL, Nicosia, Cyprus) operating at 60 kV.

### 4.4. RNA Isolation, lncRNA and mRNA Library Construction

According to the previous method [52], total RNA was isolated from each leaf of trifoliate orange sample (three biological replicates per genotype of different treatments) using the RNAprep Pure Plant Kit Polysaccharides & Polyphenolics-rich (TIANGEN Biotech, Beijing, China). TRibosomal RNA was removed using the Epicentre Ribo-Zero Gold Kit (Epicentre, Madison, WI, USA). Subsequently, sequencing libraries were generated following manufacturer recommendations with varied index labels by the NEBNext^®^ Ultra™ Directional RNA Library Prep Kit for Illumina (NEB, Ipswich, MA, USA). The libraries were sequenced on the Illumina HiSeq X Ten platform, and 150 bp paired-end reads were produced. We processed raw data by removing the adaptor polluted reads, removing the low-quality reads and trimming the reads whose number of N bases accounted for more than 5% (quality score, Q ≥ 30). In addition, the Q20, Q30 and GC content of the sequences were calculated. All downstream analyses are based on high-quality filtered sequences. All bio-informatics analyses were based on high-quality clean reads. Genes and lncRNAs were predicted based on the reference clementine mandarin (*Citrus clementina*) genome and the annotation files were downloaded from the Phytozome database (https://phytozome.jgi.doe.gov). All RNA-seq data from this work have been submitted to National Center of Biotechnology Information (NCBI) under the Sequence Read Archive (SRA) accession number PRJNA576788.

### 4.5. Expression Analysis of lncRNAs and mRNA

To measure the expression levels of lncRNAs and their targets in the leaf of trifoliate orange under B deficient stress, Cufflinks 2.1.1 was used to normalize counts of reads based on their lengths. Fragments Per Kilobase of exon model per Million mapped reads (FPKM) represented the normalized expression level of lncRNAs and target genes, and the significance of the expressional difference was tested by using the *t*-test. The fold change (FC) was measured by the DEGseq package with *p* values < 0.05, FDR (False Discovery Rate) ≤ 0.01 and |log_2_FC| > 1. Three biological replicates were performed in this work. And then the expressions of lncRNAs and mRNAs with FDR ≤ 0.01 and |log_2_FC(BD/CK)| ≥ 2 were identified as DEMs and DELs. Volcano Plot filtering was also used to identify the DEMs and DELs that reached the level of statistical significance. The heatmap was developed according to the expression abundance of lncRNAs by using the software HemI 1.0 (Heatmap Illustrator,Wuhan, China, http://hemi.biocuckoo.org/contact.php).

### 4.6. GO and KEGG Pathway Analysis

GO analysis of DEMs and target genes of DELs was performed to characterize genes and gene products in terms of cellular components, molecular functions, and biological processes. GO terms *p* < 0.05 was set as the cutoff for selecting significantly enriched functional GO terms. 

KEGG enrichment analysis was used to understand the advanced features of biological systems and the utility of the database resources (http://www.genome.jp/kegg/). In this work, a value of *p* < 0.05 was the threshold of the KEGG pathway.

### 4.7. Construction of lncRNA-mRNA Co-Expression Network

To associate the lncRNAs with direct regulated expression of target mRNAs, we superimposed lncRNA target predictions onto the lncRNA-mRNA correlation network. The resulting network was defined as an lncRNA-mRNA regulatory network. A direct connection between an lncRNA and an mRNA was represented as a solid line. Pearson’s correlation coefficient (PCC) was used to depict the co-expression relationship between lncRNA and mRNA according to their expression levels. lncRNA-mRNA pairs with |PCC value ≥ 0.90| and *p* < 0.05 were retained for network construction, which was deciphered by Cytoscape 3.6.1 (The Cytoscape Consortium, New York, NY, USA, http://cytoscape.org/).

### 4.8. Quantitative RT-PCR

Total RNA isolation from the leaf of trifoliate orange and reverse transcription for first strand cDNA were followed the method of Zhou et al. [12]. Primer pairs were designed with the Primer Express software (Applied Biosystems, Foster city, CA, USA). Primer sequences are provided in Appendix A. qRT-PCR verification was performed according to Chen et al. [25]. Technical triplicates were done for each assay, and at least three independent biological replicates were assayed. All reactions were performed in triplicate, and controls (no template and no RT) were included for each mRNA and lncRNA. The 2^−△△C*t*^ method was used to calculate the relative gene expression values.

### 4.9. Statistical Analysis

The values of plant B concentration and lignin content were presented as the means ± SE of nine samples. The data were subjected to analysis of variance (ANOVA) using SAS (SAS 8.1, SAS Institute Inc., Cary, NC, USA), and the differences were compared using the Duncan’s test at a significance level of *p* < 0.05.

## 5. Conclusions

In this study, to understand the role of lncRNA in response to B deficiency stress, and in symptom formation caused by B deficiency in the leaf of trifoliate orange, we identified and characterized its lncRNAs using RNA-seq. A total of 2,101 unique lncRNAs and 24,534 mRNAs were predicted. After identifying the expression profiling of abnormally expressed lncRNAs and protein-coding RNAs in the leaf of trifoliate orange under B deficient stress, we achieved 729 up-regulated and 721 down-regulated lncRNAs, 8419 up-regulated and 8395 down-regulated mRNAs. qRT- PCR confirmed that all selected lncRNAs responded to B deficient stress, which is consistent with RNA-seq. Further analysis showed that a total of 84 DELs were up-regulated and 31 were down-regulated, the number of DELs was 2.71-fold higher in up-regulated DELs relative to down-regulated DELs. A similar trend was also found in DEM (4.21-fold). Most of those DELs were first reported to be involved in B deficiency in plants. GO and KEGG pathway enrichment of DEMs showed that many important pathways were significantly enriched, such as biosynthesis of secondary metabolites, plant hormone signal transduction, phenylpropanoid biosynthesis and the calcium signaling pathway. Co-expression network analysis indicated that 468 significantly differentially expressed genes were regarded as potential targets for the 90 lncRNAs, and a total of 838 matched lncRNA-mRNA pairs were found. GO enrichment analysis on DELs targeting genes under B deficiency stress indicated that many lncRNAs are involved in plant hormone biosynthesis and signal transduction. In conclusion, in this work we investigated the lncRNA response to B deficiency stress, and many important lncRNAs and their target genes were found. These results in this study provide evidences for better understanding the role of lncRNAs in response to B deficiency stress and in symptom formation caused by B deficiency in the leaves of trifoliate orange.

## Figures and Tables

**Figure 1 ijms-20-05419-f001:**
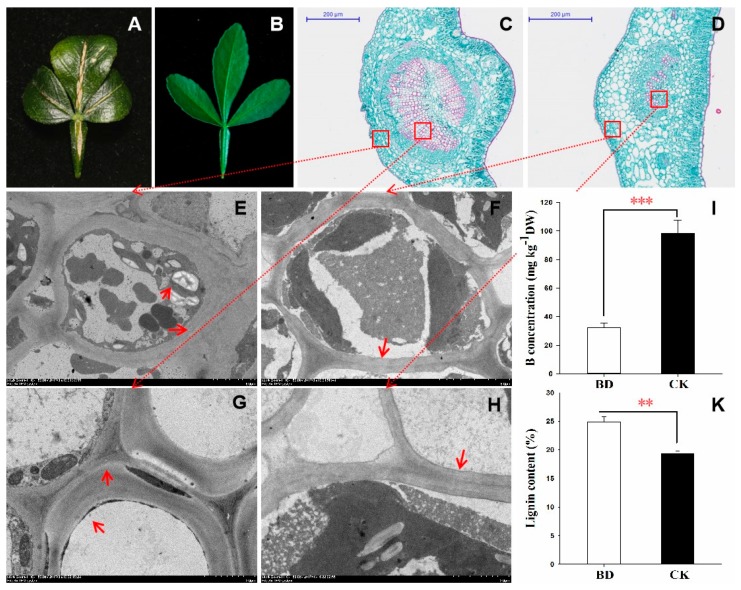
The different performance of trifoliate orange seedlings under boron deficient stress for 12 weeks. Two-month-old plants were grown in hydroponics and treated for another 12 weeks. Morphological performance of BD (**A**) and CK (**B**) leaf; Paraffin slice micrograph of the malformed vein in BD (**C**) and CK (**D**); Electron microscope observation in the cell and vessel of BD (**E**,**G**) and CK (**F**,**H**); Boron concentration (**I**) and the lignin content (**K**) in the leaf of trifoliate orange; Significant level ** *p* < 0.01,*** *p* < 0.001. Bars = 200 µm in (**C**,**D**), 10µm in (**E**,**F**), and 2µm in (**G**,**H**). BD: Boron deficiency; CK: Control.

**Figure 2 ijms-20-05419-f002:**
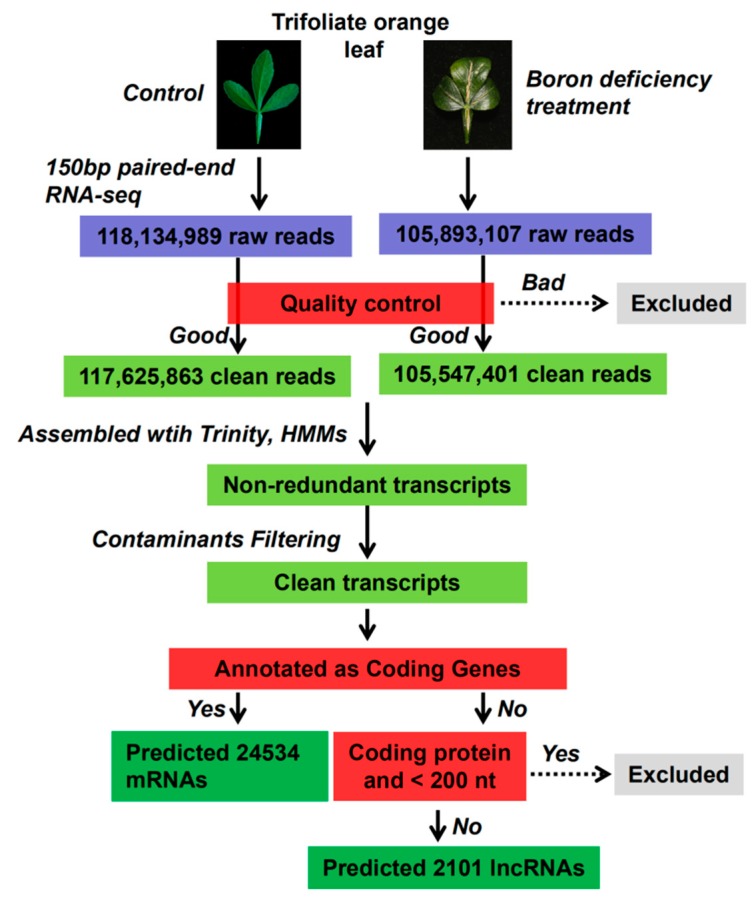
The general flowchart of the pipeline used to identify the repertoire of long non-coding RNAs (lncRNAs) from trifoliate orange expressed under boron deficient stress conditions.

**Figure 3 ijms-20-05419-f003:**
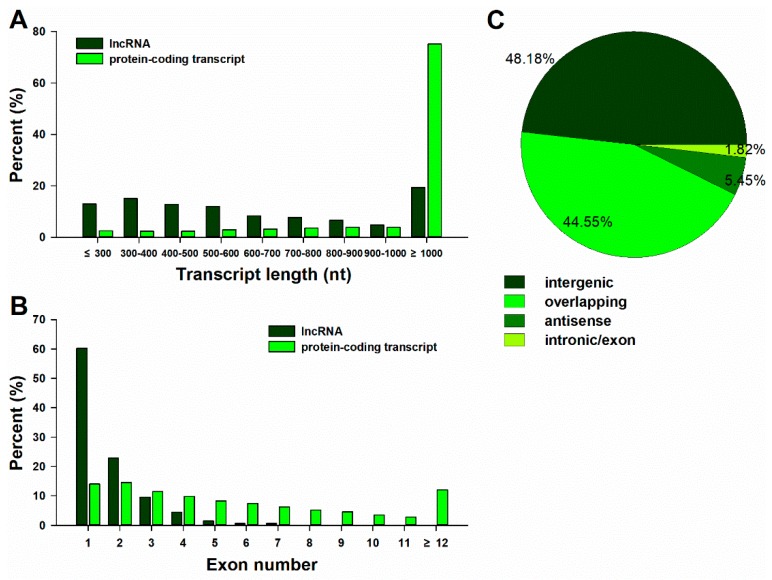
The distribution of the length (**A**) and number of exons (**B**) of lncRNAs in comparison with protein-coding transcripts and classification of lncRNAs (**C**). Protein-coding transcripts according to the Clementine mandarin (*Citrus clementina*) reference sequence. Different types of lncRNAs in the leaf of trifoliate orange, including ‘intergenic’ contained the intergenic lncRNAs, ‘overlapping’ contained the lncRNAs that have generic exonic overlap with a known transcript, ‘antisense’ contained the lncRNAs that have exonic overlap with a known transcript, but on the opposite strand, and ‘intronic/exon’ contained the lncRNAs that have intronic and exon regions of a known transcript.

**Figure 4 ijms-20-05419-f004:**
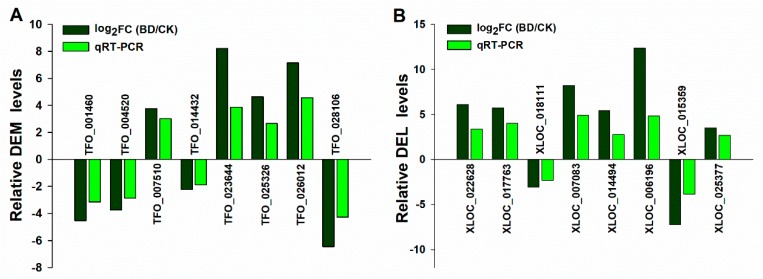
Quantitative real-time polymerase chain reaction (qRT-PCR) validation of expression levels of differentially expressed lncRNA (DELs) and differentially expressed mRNA (DEMs) in the leaf of trifoliate orange under boron deficient stress. (**A**) qRT-PCR validation of expression levels of candidate DEMs in the leaf of trifoliate orange under boron deficient stress. (**B**) qRT-PCR validation of expression levels of candidate DELs in the leaf of trifoliate orange under boron deficient stress. BD: Boron deficiency; CK: Control. FC: Fold change.

**Figure 5 ijms-20-05419-f005:**
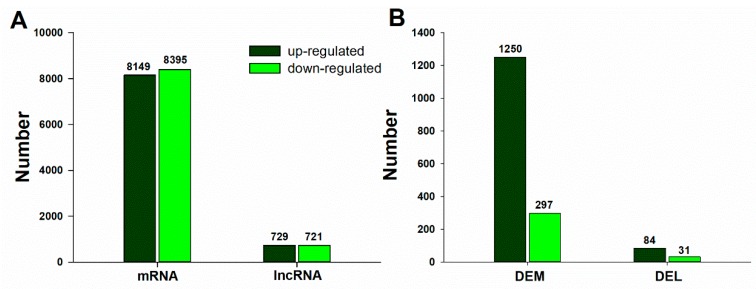
Differentially expressed lncRNAs and mRNAs in the leaves of a trifoliate orange seedling under B deficiency stress. (**A**) Total differentially expressed mRNAs and lncRNAs; (**B**) Significantly differentially expressed mRNAs and lncRNAs (|log_2_FC(BD/CK)| ≥ 2 as cutoff).

**Figure 6 ijms-20-05419-f006:**
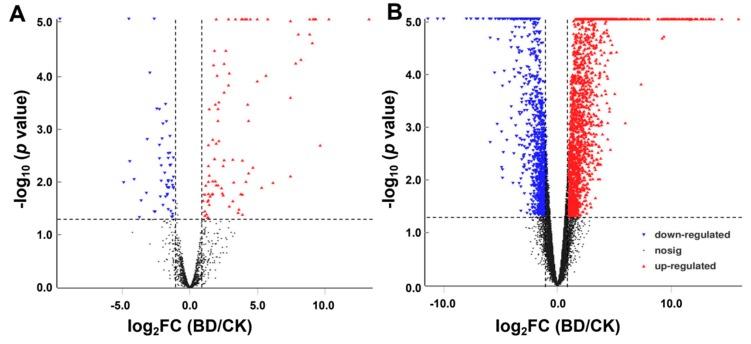
lncRNAs and mRNAs profile comparisons between BD and CK. Volcano plots were used to distinguish the differentially expressed lncRNAs (**A**) and mRNAs (**B**). The vertical lines correspond to 4-fold up-regulation or down-regulation, and the horizontal line represent *p* =0.05. The red triangle and green triangle highlight the up-regulated and down-regulated lncRNAs or mRNAs, respectively. BD: Boron deficiency; CK: Control.

**Figure 7 ijms-20-05419-f007:**
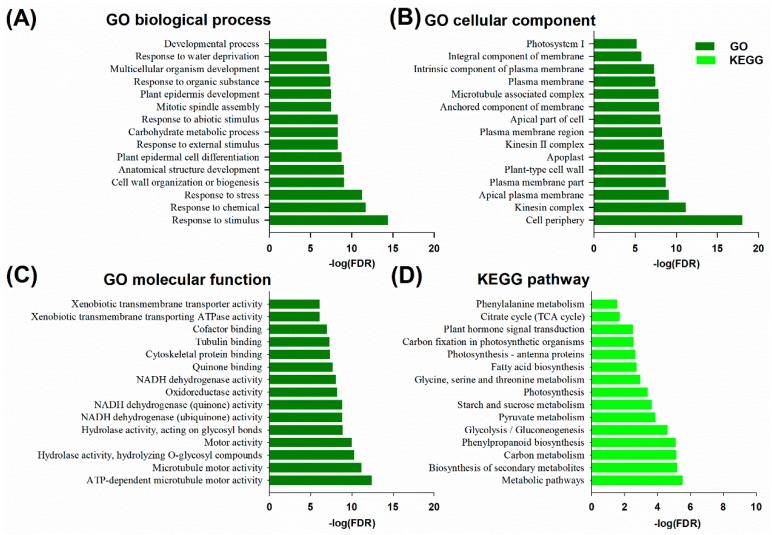
The Gene Ontology (GO) terms and Kyoto Encyclopedia of Genes and Genomes (KEGG) pathway enrichment of differentially expressed mRNAs (DEMs) in the leaves of trifoliate orange subjected to boron deficiency stress. GO stands for gene ontology; FDR for false discovery rate; KEGG, the Kyoto encyclopedia of genes and genomes. (**A**) Biological process of GO terms. (**B**) Cellular component of GO terms. (**C**) Molecular function of GO terms. (**D**) KEGG pathways.

**Figure 8 ijms-20-05419-f008:**
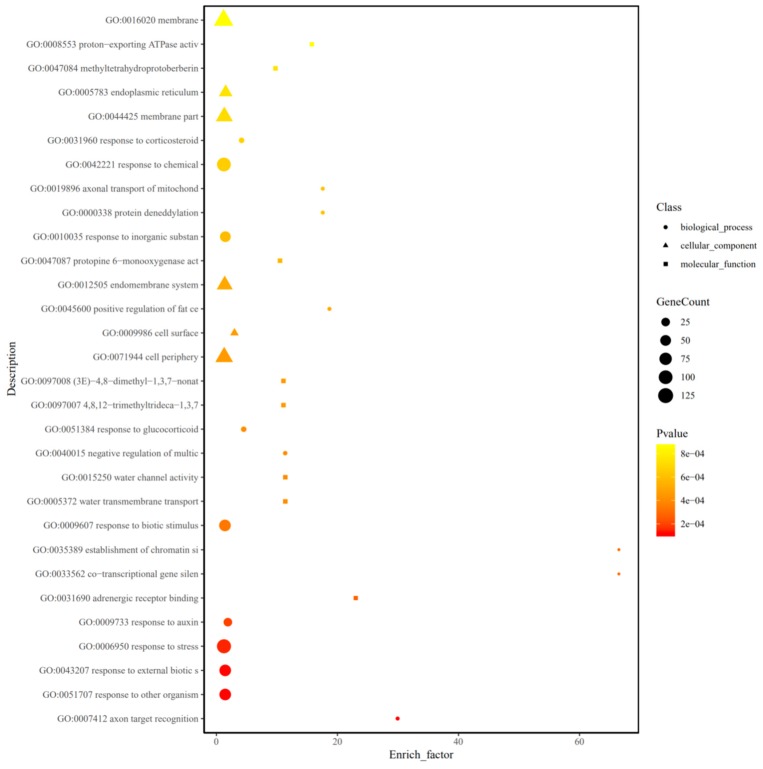
Gene ontology (GO) enrichment analysis on target genes of differentially expressed lncRNAs under boron deficiency stress. The GO analysis and enrichment is measured by enrich factor, *P* value, and the number of genes enriched on this pathway (Enrich_factor = GeneRatio/BgRatio). The larger the enrich factor, the greater the degree of enrichment. The larger the graph (circle: Biological process; triangle: Molecular function; square: Cellular component), the greater the number of differentially expressed genes. The darker the color, the more significant the enrichment of the GO term.

**Figure 9 ijms-20-05419-f009:**
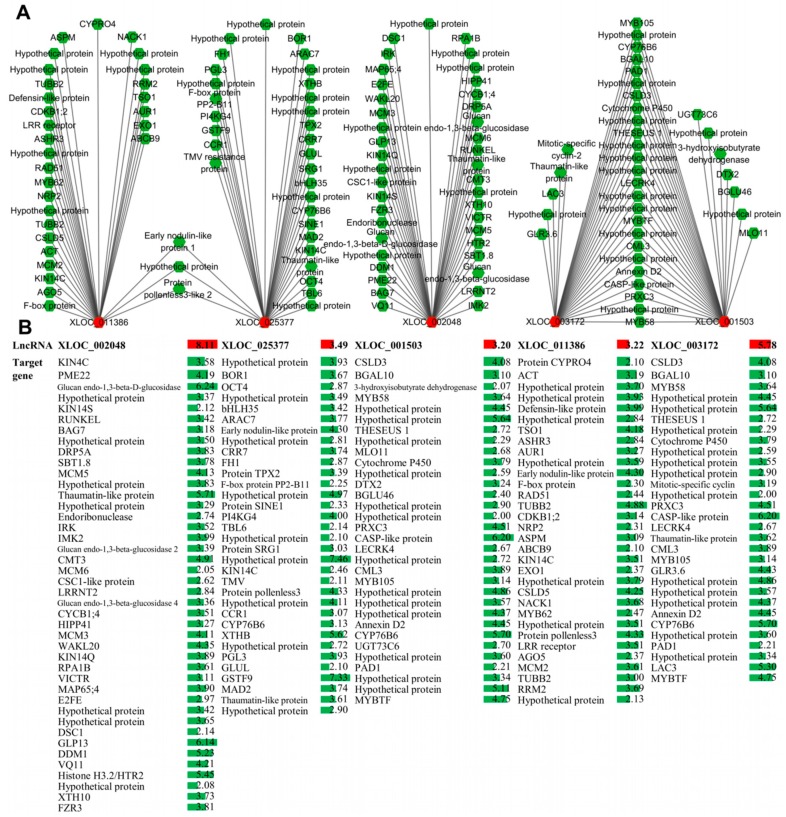
The top five lncRNAs of the interaction network among lncRNAs and target protein-coding genes. (**A**) Co-expression network. The nodes represent lncRNAs and protein-coding genes and edges show regulatory interactions among nodes. (**B**) Data bars with different colors show the expression levels of lncRNAs (colored in red) and target protein-coding genes (colored in green) under boron deficiency conditions.

**Figure 10 ijms-20-05419-f010:**
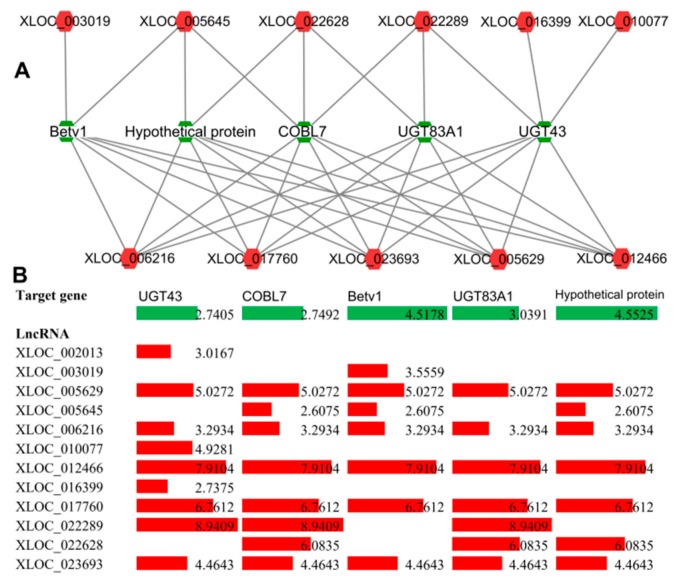
The top five target genes of the interaction network among lncRNAs and target protein-coding genes. (**A**) Co-expression network. The circle nodes represent lncRNAs and protein-coding genes, and edges show regulatory interactions among nodes. (**B**) Data bars with different colors show the expression levels of lncRNAs (colored in red) and target protein-coding genes (colored in green) under boron deficiency conditions.

**Figure 11 ijms-20-05419-f011:**
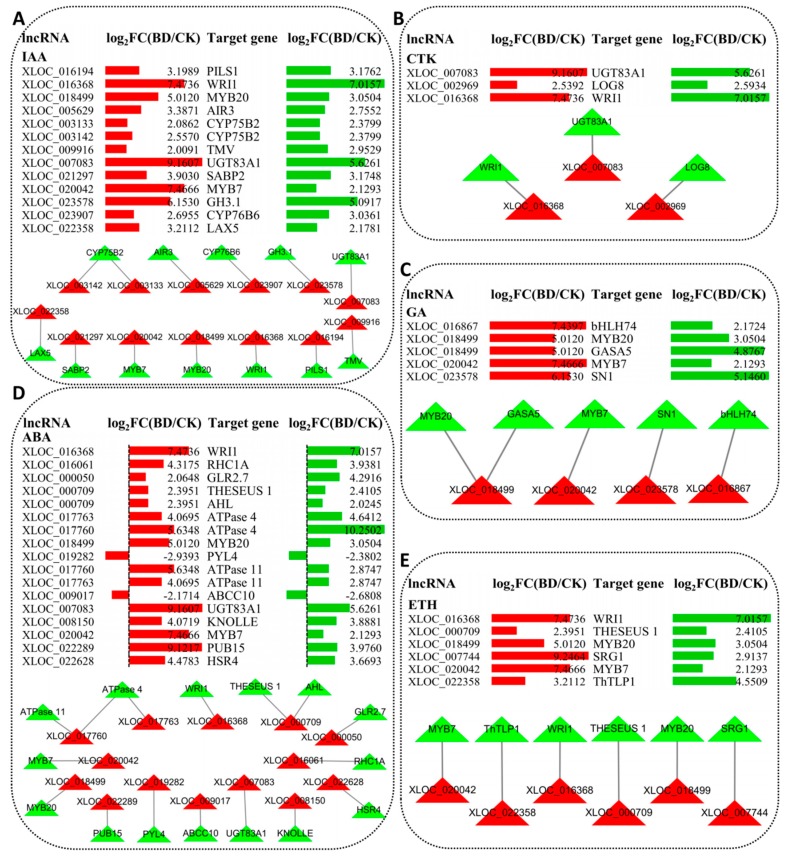
LncRNAs involved in plant hormones biosynthesis and signal transduction. (**A**) Auxin (IAA) relative lncRNAs and their target genes; (**B**) Cytokinin (CTK) relative lncRNAs and their target genes; (**C**) Gibberellin acid (GA) relative lncRNAs and their target genes; (**D**) Abscisic acid (ABA) relative lncRNAs and their target genes; (**E**) Ethylene (ETH) relative lncRNAs and their target genes. Data Bars with different colors showed the expressions of lncRNAs (colored in red) and target protein-coding genes (colored in green) under boron deficiency conditions, respectively. For the co-expression network, the rectangle nodes represent lncRNAs (colored in red) and protein-coding genes (colored in green), and edges show regulatory interactions among nodes.

**Table 1 ijms-20-05419-t001:** Summary of the RNA-seq data for six samples.

Summary	Control			Boron Deficiency		
	CK_1	CK_2	CK_3	BD_1	BD_2	BD_3
Raw reads	106,350,380	128,309,312	119,745,274	113,804,668	100,898,426	102,976,228
Clean reads	105,992,604	127,931,954	118,953,032	113,440,572	100,542,180	102,659,450
Clean reads rate (%)	99.66	99.71	99.34	99.68	99.65	99.69
Q20%	97.81	97.83	97.62	97.71	97.70	97.82
Q30%	94.43	94.44	93.98	94.16	92.20	94.43
GC%	42.24	42.57	42.66	42.75	43.08	43.03
Mapped reads	30,810,136			30,324,690		
Unique mapped reads	22,371,249			25,678,257		

Q20% represents the proportion of the data in which the quality values are > Q20 in the raw data. Q30% represents the proportion of the data in which the quality values are > Q30 in the raw data. BD: Boron deficiency treatment; CK: Control.

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
