# Peer review of "Genome-Wide Identification of Long Non-coding RNA in Trifoliate Orange (Poncirus trifoliata (L.) Raf) Leaves in Response to Boron Deficiency"

_ijms, 2019, doi:10.3390/ijms20215419_

Round 1

Reviewer 1 Report

The manuscript “Genome-wide Identification and Characterization of Long Non-coding RNA in Trifoliate Orange [Poncirus trifoliata (L.) Raf] Leaves in Response to Boron Deficiency” describes LncRNAs in trifoliate orange. The manuscript itself is descriptive and there is no characterization as such. Therefore, I recommend removing the “characterization” word from the title. Even though the amount of analysis presented in the manuscript is interesting, it suffers from a number of problems that need to be fixed. These problems are outlined below. 

In general, the manuscript is well written, however, there are some English errors (grammar, usage, syntax) that need to be fixed. For instance, page 2; lines 92 and 93. “Vein swelling and cracking, typical symptoms of citrus have been proved in various citrus, were observed in the old leaves of BD”

Was the RNA-Seq performed on the entire shoots? A few leaves? Top leaves? It is not clear from the material and methods section what exact leaf material was used.

Figure 8 is not informative. The authors describe the top 5 lncRNA interaction network with their targets. The Figure should be visually improved. The black box in the background and the small letters do not make it appealing. There is no relevant information of them in the result section. What do those networks tell us about boron deficiency? In figure 8B, the authors need to realize that the publications can be accesed by non-expert and expert readers as wells as ones not familiar with trifoliate orange and lncRNA, the CiclevXXXX identifiers do not mean anything. Provide the gene function of the targets, so readers can make an association, otherwise, it is not adding to the manuscript. Similar for figure 9 and 10.

GO classification should be used as a step to move deeper in the analysis not as a result itself. The authors need to explore further and improve their analysis.

The results section is not clear. The authors superficially describe some of the results leaving the impression of lack of connection between sections.

The way the qRT-PCR validation section in the results is written does not add anything to the manuscript. What else could you comment besides that the results match the RNA-Seq?

Similarly, for section 2.6 LncRNA-mRNA interaction.

In figure 11, do the validated lncRNA have a number assigned? In that way, further studies could easily track and compare the findings of this study.

It is interesting and I am extremely curios about the fact that the authors completely ignored a paper from 2017 entitled “Genome-wide screening and characterization of long non-coding RNAs involved in flowering development of trifoliate orange (Poncirus trifoliata Raf.)”. That manuscript could give the authors the opportunity to compare and explore whether lncRNA identified by that study could relate to boron deficiency. Dual function? Novel lncRNA?

RNA-Seq data needs to be deposited in the NCBI repository. Provide accession numbers.

The discussion section needs to be improved.

Reviewer 2 Report

The manuscript entitled "Genome-wide Identification and Characterization of Long Non-coding RNA in Trifoliate Orange [Poncirus trifoliata (L.) Raf] Leaves in Response to Boron Deficiency" provided an inventory of lncRNAs in trifoliate orange against Boron deficiency stress. 

It is very hard nowadays to find a research paper that, on average, is sound and complete in terms of experimentation, statistical analysis, and particularly write-up. This manuscript has been prepared in such a way that it leaves a good impression on the reviewer. I have read it multiple times and did not find any discrepancies. Hence, I have no doubt in recommending this article for publication.

Minor Comment:

P2L92: “were observed in the old leaves of BD”, authors are advised to give precise information about days/weeks after which the symptoms were observed.

Author Response

Thank you very much for your critical review and kind comments on our manuscript. Based on the suggestions from the editor and reviewers, the manuscript was accordingly revised and marked red in the text. We hope that the revised version could meet the requirements for publication. Here, the responses to the comments were listed as below:

Point 1: P2L92: “were observed in the old leaves of BD”, authors are advised to give precise information about days/weeks after which the symptoms were observed.

Response 1: Thanks for your suggestions. We are very sorry for our negligence of these errors (and grammar, usage, syntax et al.), and they have been corrected in revised manuscript.

We appreciate for your warm work earnestly, and hope that the correction will meet with approval. Once again, thank you very much for your comments and suggestions.

Round 2

Reviewer 1 Report

The Authors clearly improved the resubmitted version, however, there are still figures difficult to read.
